climate change; eco-anxiety; extreme weather; air pollution; mental health; suicide risk and suicidality

**Corresponding author:**
Darya Rostam Ahmed;
Emails: darya.rostam@koyauniversity.org

# A systematic review of the association between climate change and suicidality reveals that climate indicators increase suicide rates

Darya Rostam Ahmed[1] 🔘, Sujita Kumar Kar[2], Mohammad Al Diab Al Azzawi[3] and Reinhard Heun[4]

[1]Koya University, Iraq; [2]King George's Medical University, India; [3]National Ribat University, Sudan and [4]University of Bonn, Germany

## Abstract

Climate change is increasingly recognized as a public health challenge, with emerging evidence linking climate-related factors to suicidality. A search was conducted in PubMed, Scopus, PsycINFO, Web of Science and Google Scholar following the PRISMA guidelines. The studies that assessed the association between climate indicators and suicidality were included, and risk of bias was assessed using MMAT and ROBINS-E. A total of 43 studies met the inclusion criteria, covering various geographic regions and populations. Rising ambient temperatures were the climate variable most frequently studied, with multiple studies showing a significant increase in suicide rates linked to higher temperatures, particularly during the summer months, especially among females. Seasonal variations, including heatwaves and extreme cold, were associated with increased suicidality. Additionally, extreme weather events such as floods, droughts and storms correlated with higher suicide risks, particularly in vulnerable populations, including older adults and individuals with pre-existing mental health conditions. Air pollution, particularly exposure to PM2.5, NO2 and SO2, was also found to contribute to suicidality. Most of the studies originated in high-income countries, highlighting a gap in research from low- and middle-income countries (LAMICs), where the impacts of climate change may be more severe but remain understudied. Although two studies examined suicidal ideation, the overwhelming majority of the evidence focused on suicide mortality, underscoring the marked under-representation of non-fatal suicidality outcomes in the existing literature. The findings suggest that climate change plays an important role in suicidality, with increasing temperatures, extreme weather and air pollution acting as key risk factors. As climate stressors grow, it is crucial to integrate them into mental health and suicide-prevention policies. More research, especially in underrepresented regions, is needed to guide effective interventions.

## Impact statements

The present review brings together global evidence showing that climate change is not only an environmental threat but also a growing driver of suicide risk. Across diverse countries and populations, the findings demonstrate that higher temperatures, heatwaves, air pollution and extreme weather events such as floods and droughts are consistently linked to increases in suicidal behaviour, particularly fatal suicide. As these climate stressors become more frequent and intense, their influence on population mental health is likely to increase further. The results show that climate-related suicide risk is not uniform across society. Older adults, people with pre-existing mental health conditions, pregnant women and individuals exposed to severe environmental stress appear especially vulnerable. Seasonal and temperature-related effects are also evident, highlighting how changes in daily living conditions can affect psychological stability and crisis risk. The review also exposes a major equity concern. Most of the existing evidence comes from high-income countries, even though many of the most severe climate shocks occur in LAMICs, where social protection systems and mental health services are often weakest. This imbalance means that the people who are likely to be most affected by climate-related distress and suicide risk are also those least represented in the scientific literature. Importantly, the review supports concrete actions that can be implemented now. These include linking heat and air-quality warning systems with mental health services, establishing cooling centres during extreme heat and integrating mental health and electronic mental health support (eMHPSS) into disaster preparedness and response. Such measures have the potential to reduce psychological distress and prevent loss of life during climate-related emergencies. By clearly demonstrating that climate change is a direct suicide-prevention issue, this paper strengthens the case for embedding mental health protection within climate adaptation, public health policy and disaster risk-reduction strategies worldwide.





## Introduction

Climate change represents one of the most significant challenges of our era, with substantial potential impacts on human beings (Rocha et al., 2022). Climate change refers to long-term changes in local, global or regional temperatures and weather caused by human activity. (Shivanna, 2022) Over the years, human life, flora and fauna have coexisted and developed within an environment that has experienced relatively stable climatic conditions, particularly in terms of temperature, humidity and solar radiation. Although there have been notable historical exceptions such as ice ages, medieval catastrophes and major natural disasters such as earthquakes and plagues, these events were either localized or occurred over extended periods, allowing some level of adaptation. As a result, the capacity of humans, plants and animals to adapt to sudden or extreme climatic changes, such as those driven by contemporary global warming, remains inherently limited. According to the results of the Lancet countdown in Europe, the impact of climate change on health is already clear throughout Europe. With the increase in frequent severe weather events, these impacts become increasingly evident, putting environmental conditions at risk from factors that cause the spread of infectious diseases and cause disruption of the water and food systems. (Romanello et al., 2021) Climate change manifests itself in various forms, including acute weather events or natural disasters that occur over a few days, such as hurricanes, wildfires, floods and short-term heatwaves. It also includes subacute events that persist for months or a few years, such as droughts (Bell et al., 2018). Furthermore, there are long-lasting changes that could extend until the end of the century, such as higher temperatures, extensive melting of glaciers, rising sea levels and the potential emergence of inhabitable physical environments due to permanent alterations (Global Change and Future Earth: The Geoscience Perspective – Google Books, n.d.). The World Health Organization (WHO) has put forward five key global research priorities to protect human health from the impact of climate change. These priorities include evaluating the risks involved, identifying the most effective interventions, providing guidance for health-promoting measures in other sectors, improving decision support systems and estimating the costs associated with protecting health from climate change (WHO, 2009).

The consequences of climate change have profound effects on individuals; people experience exposures and events that include mental health consequences that range from relatively mild stress symptoms to diagnosed psychiatric disorders. (Ebi et al., 2018; Hayhoe et al., 2018) For example, prolonged increases in average temperature have been associated with serious adverse outcomes, including higher rates of suicide, increased violence or aggression and poorer self-reported mental health. (Basu et al., 2018; Burke et al., 2018; Mullins and White, 2019) Extreme weather events are associated with increased frequency and duration, including heatwaves, storms, droughts, wildfires, heavy rainfall, floods, tropical storms and storm surges, and these combined or independent events are associated with post-traumatic stress disorder (PTSD), depression, mood disorders, anxiety and an increased risk of violence. (Vins et al., 2015; Dodgen et al., 2016) Extreme weather can also contribute to disturbing activities that people display outside to relieve stress. Also, such conditions lead people to stay inside, causing even more mental health problems for individuals (Dodgen et al., 2016).

Despite growing evidence linking climate indicators such as rising temperatures, extreme weather events and air pollution to mental health problems, including suicidality, existing research remains fragmented. In the present review, suicidality is used as an umbrella term encompassing a spectrum of suicide-related outcomes, including suicidal ideation (thoughts of ending one's life), suicide attempts (non-fatal, self-injurious behaviour with intent to die) and suicide completion (death by suicide). Most studies originate from high-income countries, while low- and middle-income countries (LAMICs) where climate stressors are often more severe remain underrepresented. Furthermore, prior studies vary widely in methodology, exposure measures and populations studied, limiting the comparability of findings. Few studies have synthesized these diverse results to provide a global perspective on how climate change influences suicide risk. Addressing this gap is essential to inform mental health policy, guide suicide prevention strategies and identify priorities for future research. Therefore, this review systematically examines the evidence on climate change and suicidality, highlighting global patterns, methodological challenges and critical gaps in knowledge.

## Method

### Study protocol and registration

The current systematic review was conducted according to the Preferred Reporting Items for Systematic Reviews and Meta-Analyses (PRISMA) checklist (Page et al., 2021a). (See Appendix). The protocol for this systematic review was registered in the International Prospective Register of Systematic Reviews (PROSPERO) with the registration number **CRD42024539305**.

### Data collection and search strategy

We conducted a comprehensive literature search across five electronic databases and search engines: PubMed, Scopus, PsycINFO, Web of Science and Google Scholar. The search strategy combined terms for climate change with terms for suicidality, using Boolean operators and truncation to capture variations. For example, the PubMed search string was: ("climate change" OR "global warming" OR "temperature" OR "heatwave" OR "air pollution" OR "extreme weather" OR "flood" OR "drought") AND ("suicide" OR "suicidal ideation" OR "suicide attempt" OR "self-harm" OR "self-injurious behavior"). The suicidality construct was operationalized broadly to include ideation, attempts and completed suicides, ensuring that studies addressing any stage of suicidality were captured. No restrictions were applied regarding study setting or population. Reference lists of included articles and relevant reviews were also manually screened to identify additional studies.

### Eligibility criteria

#### Inclusion criteria

The inclusion criteria were: (1) peer-reviewed studies using quantitative data that examined the link between climate change and suicidality, (2) studies focusing on how climate indicators such as rising temperatures and extreme weather events influence the risk of suicide, (3) observational studies and randomized controlled trials and (4) studies published in English.

#### The exclusion criteria

Exclusion criteria were non-peer-reviewed studies, conference abstracts, editorials, reviews, studies not related to climate change

and suicide rates, studies focused solely on mental health disorders without addressing suicide, qualitative studies without quantitative data and studies published in languages other than English. During screening, a small number of qualitative papers were identified; however, these were largely narrative in nature, discussing the general impact of climate change on mental health without providing empirical data, statistical testing or suicide-specific outcomes. The focus on quantitative studies was chosen to allow for the assessment of measurable associations between climate indicators and suicidality outcomes using effect size estimates and statistical parameters that could be meaningfully compared across studies. Mixed-methods studies were considered eligible only when the quantitative component could be clearly extracted and independently analysed; however, none met this criterion during full-text screening. While qualitative research provides valuable contextual and experiential insights, it was excluded because it does not permit quantitative synthesis of risk or exposure effects.

### Study screening

The study screening process was carried out by two independent researchers (S.K.K and M.D.A), who reviewed the titles and abstracts of the identified articles for eligibility using the Rayyan AI platform (Ouzzani et al., 2016), which helped streamline the screening process by allowing efficient sorting of articles based on predefined inclusion and exclusion criteria. They then performed a full-text screening based on these criteria. Subsequently, the lead author (D.R.A) performed a secondary screening to ensure that all studies met the inclusion criteria. Disagreements between the two screeners were minimal and were resolved through discussion with the lead author (D.R.A), ensuring consistency and accuracy in the study selection process.

### Data extraction

Data extraction was conducted using a standardized form that captured key study characteristics, including the first author's name, publication year, study country, study period, design, population, sample size, gender, age, climate change and suicide measurements, type of climate exposure and main findings. The detailed characteristics of the included studies are presented in Table 1.

### Assessment of risk of bias

The risk of bias assessment was performed based on the included research design; we applied two different tools for the quality assessment of the included articles in the present review. We used the Mixed Methods Appraisal Tool (MMAT) (Hong et al., 2019) because there are studies with different designs and no exposure. We evaluated the research question, study design, sample size, data collection and statistical analysis using MMAT. We also used the ROBINS-E (risk of bias in nonrandomized studies of exposures) tool (Higgins et al., 2024) because it is specifically designed to assess the risk of bias in nonrandomized observational studies that evaluate exposures rather than interventions. Observational studies often face unique challenges compared to randomized controlled trials (RCTs), such as confounding, selection bias and measurement errors.

### Data synthesis and analysis

We performed a systematic narrative analysis, which allowed us to summarize and interpret the data while identifying patterns, trends and recurring themes in the relationship between climate variables and suicidality. A meta-analysis was not performed due to substantial heterogeneity in study designs, measurement approaches, populations, climate exposures and suicidality outcomes, which precluded meaningful pooling of effect sizes. Narrative synthesis was therefore deemed the most appropriate approach to integrate the findings across diverse methodologies and contexts.

## Results

### Study selection process

A total of 748 articles were identified through a comprehensive search, including PubMed, Scopus, PsycINFO, Web of Science and Google Scholar. After removing 343 duplicates, the remaining 405 articles underwent title and abstract screening, leading to the exclusion of 286 articles that did not meet the inclusion criteria for example, studies focusing on climate change and general mental health outcomes without reporting suicidality, or studies limited to physical health impacts. The full texts of the remaining 119 articles were reviewed, resulting in the exclusion of 76 articles due to ineligible criteria (49), assessment of psychosocial aspects without suicide (17), insufficient data (7) and narrative studies (3). Ultimately, 43 articles met the inclusion criteria and were included in the review, as presented in the PRISMA flow diagram (Figure 1).

### Study characteristics

The included studies came from a range of countries and regions. The United States contributed seven studies (Bakian et al., 2015b; Burke et al., 2018; Bergmans et al., 2021; Cheng et al., 2021; Rahman et al., 2023; Freichel & O'Shea, 2023; Runkle et al., 2024), Italy five (Preti, 1997; 1998 ; Preti et al., 2007; Rocchi et al., 2007; Di Nicola et al., 2020; Aguglia et al., 2021) Turkey four (Doganay et al., 2003; Akkaya-Kalayci et al., 2017; Asirdizer et al., 2018; Kayipmaz et al., 2020), Australia three (Matthews et al., 2019; Edwards et al., 2023; Hertzog et al., 2024), Germany three (Müller et al., 2011; Schneider et al., 2020; Brailovskaia and Teismann, 2024), Finland two (Helama et al., 2013; Hiltunen et al., 2014), Canada two (Villeneuve et al., 2023; Freichel & O'Shea, 2023) and the rest from other European, Asian and Latin American countries. Most of the included articles were retrospective in design, with a few exceptions where case-crossover (Bakian et al., 2015a; Lee et al., 2019; Bergmans et al., 2021; Rahman et al., 2023; Villeneuve et al., 2023; Lian et al., 2024; Runkle et al., 2024), cross-sectional (Matthews et al., 2019; Di Nicola et al., 2020; Miyazaki et al., 2023; Ndetei et al., 2024), case control (Makris et al., 2021) and descriptive survey (Kayipmaz et al., 2020) designs were used. Although several included articles were published in 2024, the study periods and datasets they analysed covered years up to 2023, with durations ranging from a few months (Edwards et al., 2023) to more than 50 years (Helama et al., 2013). Participants were exposed to a varying number of climate types, including extreme weather, high temperature, rainfall, air pollutants, flooding and humidity with suicidal behaviour ranging from mere ideation to complete/fatal suicide. The detailed characteristics of the included studies are presented in Table 1.

### Participants

Studies covered a diverse population, with the general population being the group most frequently examined, followed by university students, employees, high school students, pregnant women, older adults (≥65 years), suicidal individuals, adolescents and those with mental illnesses such as bipolar disorder. Sample sizes varied

**Table 1.** Characteristics of the included studies, measurements of climate and suicide, types of climate exposure and main findings

| References | Country | Study period | Study design | Population | Sample size and gender | Age | Climate measurement | Suicide measurement | Climate exposure/ climate types | Finding |
|---|---|---|---|---|---|---|---|---|---|---|
| Brailovskaia and Teismann, 2024 | Germany | 2023 | Cross-sectional | University Students and employees | N: 323 ♀: 221 ♂: 102 | 18–63 | Climate Change Stress and Impairment Scale (CC-DIS) | Suicide Ideation and Behavior Scale (SIBS) | Eco-anxiety | Climate change anxiety/impairment was significantly positively associated with suicidal ideation. |
| Hertzog et al., 2024 | Australia | 2000–2019 | Case-crossover | General population | N: 50,733 ♀: 11,932 ♂: 38,802 | 0–55+ | Retrospective statistics about climate | Retrospective statistics about suicide | Temperature/heat anomaly | Suicides were associated with climate change-induced heat anomalies, with a significant association observed among men 55 and older. |
| Lian et al., 2024 | China | 2014–2020 | Case-crossover | General population | N: 30,688 | Not provided | Retrospective statistics about climate | Retrospective statistics about suicide | Air pollution | Short-term exposure to air pollution, particularly in cities at the county level, was associated with increased suicide deaths. The pollution mixture was associated with an increase in suicide mortality, potentially contributing to up to 8.07% of excess suicide deaths. |
| Ndetei et al., 2024 | Kenya | 2022 | Cross-sectional | High school students | N: 2,652 | 13–20 | A survey without specifying the tool | A survey without specifying the tool | Perceived impact of climate change | Suicidal thoughts and attempts linked to perceived threats of climate change. |
| Runkle et al., 2024 | USA | 2016–2019 | Case-crossover | Pregnant women | N: 1,491 ♀: 1,491 | 18–35+ | Retrospective statistics about climate | Retrospective statistics about suicide | Warm ambient temperature | Each 5 ° C increase in exposure to warm ambient temperature on case days was associated with a significant increase in suicidal thoughts (IRR = 1.28; 95% CI: 1.00, 1.65). |
| Edwards et al., 2023 | Australia | 2000–2004 | Longitudinal | Adolescents | N: 2,908 | 14–19 | Retrospective statistics about climate disasters | Retrospective statistics about suicide | Natural disasters (fire/flood & drought) | Natural disasters, such as fires, floods and droughts, are associated with increased risk of suicidal thoughts and behaviours. |
| Freichel & O'Shea 2023 | USA, UK, and Canada | 2012–2018 | Retrospective | General population | N: 10,448 | Not provided | Not provided | Self-harm implicit association test (IAT) | Seasonality | There is an association between climate and suicide. Negative thoughts of self-harm, including suicidal ideation, peaked in winter (December), with implicit thoughts reaching a delayed peak in February. |
| Miyazaki et al., 2023 | Japan | 2005–2022 | Cross-sectional | Individuals with suicidal behaviour | N: 1,737 | Not provided | Retrospective statistics about air pollution | Retrospective statistics about suicide | Air pollution | A significant association between air pollution, specifically SO2 levels, and increased suicide attempts, while NO (nitrogen oxide) levels were negatively related to suicide attempts. |
| Rahman et al., 2023 | USA | 2014–2019 | Case-crossover | General population | N: 24,387 | Not provided | Retrospective statistics about air pollution | Retrospective statistics about suicide | Ambient temperature & air pollution | Increased ambient temperature was associated with increased odds of suicide deaths. However, no associations with suicide were observed for air pollutants. |

(*Continued*)

Cambridge Prisms: Global Mental Health

| References | Country | Study period | Study design | Population | Sample size and gender | Age | Climate measurement | Suicide measurement | Climate exposure/ climate types | Finding |
|---|---|---|---|---|---|---|---|---|---|---|
| Villeneuve et al., 2023 | Canada | 2002–2015 | Case-crossover | General population | N: 50,800 | Not provided | Retrospective statistics about air pollution | Retrospective statistics about suicide | Ambient temperature & air pollution | Increases in NO2, PM2.5 and O3 were associated with increased suicide mortality, with the strongest associations in women and during warmer seasons. Temperature increases also increased the risk of suicide, amplified by higher levels of PM2.5 and O3. |
| Jenwitheesuk et al., 2022 | Thailand | 2010–2017 | Retrospective | Individuals with suicidal behaviour | N: 4,756 ♀: 934 ♂: 3,822 | Not provided | Retrospective statistics about air pollution | Retrospective statistics about suicide | Air pollution | Each 1 mg/m³ increase in dust-PM2.5 was significantly associated with a 63.5% increase in suicide risk, while black carbon, organic carbon and sulphate did not show significant effects. |
| Aguglia et al., 2021 | Italy | 2016–2018 | Retrospective | Individuals with suicidal behaviour | N: 225 ♀: 170 ♂: 55 | 40–60 | Retrospective statistics about air pollution | Retrospective statistics about suicide | Solar radiation & air pollution | The study found a significant association between apparent temperature and suicide, with solar radiation (65%) and PM2.5 (32%) positively correlated. The male gender and a phase change of <1 month also showed strong links. |
| Bauer et al., 2021 | International study, 40 countries | 2010–2020 | Retrospective | Bipolar patients | N: 4,876 ♀: 2,760 ♂: 2,116 | Adults | Retrospective statistics about climate | Retrospective statistics about suicide | Seasonal solar insolation | Large changes in solar insolation between seasons or monthly extremes may increase the risk of suicide attempts in individuals with bipolar I disorder. |
| Bergmans et al., 2021 | USA | 2007–2015 | Case-crossover | Individuals with suicidal behaviour | N: 12,646 ♀: 2,630 ♂: 10,016 | M: 46.3 | Retrospective statistics about climate | Retrospective statistics about suicide | Aeroallergen | No general association was found between aeroallergens and suicide, but grass pollen was associated with suicide in women and individuals with a high school education or less. |
| Cheng et al., 2021 | USA | 1999–2019 | Retrospective | General population | N: 38,000 | Not provided | Retrospective statistics about climate | Retrospective statistics about suicide | Increased temperature | A 1 °C increase in average temperature was associated with a 0.82% increase in suicide rates, with similar effects observed for maximum and minimum temperatures. |
| Makris et al., 2021 | Sweden | 2006–2012 | Case–control | Individuals with suicidal behaviour | N: 1,981 | M: 56.1 | Retrospective statistics about climate | Retrospective statistics about suicide | Temperature/ sunshine | Among older adults (65+), higher temperatures and increased sunshine were associated with a slight increase in suicide rates and attempts, although these associations were not significant after adjustments. |
| Bozsonyi et al., 2020 | Hungary | 1971–2013 | Retrospective | General population | N: 91,509 | Not provided | Retrospective statistics about climate | Retrospective statistics about suicide | Sunshine duration & ambient temperature | Consistent and immediate positive associations were observed between daily suicide rates, elevated ambient temperature and duration of sunshine in a high-rated area of Hungary. |

(Continued)

**Table 1.** (*Continued*)

| References | Country | Study period | Study design | Population | Sample size and gender | Age | Climate measurement | Suicide measurement | Climate exposure/ climate types | Finding |
|---|---|---|---|---|---|---|---|---|---|---|
| Schneider et al., 2020 | Germany | 1990–2006 | Case-crossover | General population | N: 10,595 ♀: 3,284 ♂: 7,311 | ≥ 65 | Retrospective statistics about climate | Retrospective statistics about suicide | Air temperature, sunshine duration & cloud cover | An increase in air temperature the day before suicide was linked to a 5.7% higher risk. Suicides were also more likely on days with low/medium cloud cover, with stronger effects in summer and fall. Temperature increases in these seasons were associated with a higher risk of suicide. |
| Kayipmaz et al., 2020 | Turkey | 2017–2019 | Retrospective | General population | N: 6,777 ♀: 3,122 ♂: 3,655 | *M:* 32.15 | Retrospective statistics about climate | Retrospective statistics about suicide | Temperatures variables (maximum, minimum humidity & actual pressure) | An increase in 1 ° C in the minimum temperature on the day of application was associated with a 0.01 increase in suicides (p = 0.046). No other significant changes were observed in the variables. |
| Nicola et al., 2020 | Italy | 2016–2019 | Cross-sectional | Bipolar patients | N: 352 ♀: 222 ♂: 130 | 18–70 | METEO-Q | ICD–10 & Interview | Seasonal & climatic-related variations | The study found a significant association between lifetime suicide attempts and sensitivity to weather and climate variations in patients with bipolar disorder (BD). Higher METEO-Q scores, indicating greater sensitivity to climate changes, were correlated with an increased number of suicide attempts. |
| Koszewska et al., 2019 | Poland | 1999–2014 | Retrospective | General population | N: 210 ♀: 42 ♂: 168 | 25–65 | Retrospective statistics about climate | Retrospective statistics about suicide | Foehn wind | The foehn wind could increase the risk of suicide mainly in summer and autumn. |
| Lee et al., 2019 | South Korea | 2002–2015 | Case-crossover | Individuals with suicidal behaviour | N: 30,704 ♀: 10,381 ♂: 20,323 | 35–65 | Retrospective statistics about climate | Retrospective statistics about suicide | Dust storm | Dust storms were associated with an increased risk of suicide, with stronger associations during prolonged and intense storms. |
| Matthews et al., 2019 | Australia | 2017 | Cross-sectional | General population | N: 2,180 ♀: 1,500 ♂: 680 | 16–75 | Flood exposure | Single item tool for assessing risk of suicide | Flood | Flood was associated with a 7% increase in suicidal ideation. |
| Asirdizera et al., 2018 | Turkey | 2006–2015 | Retrospective | General population | N: 29,865 ♀: 8,845 ♂: 21,020 | Not provided | Retrospective statistics about climate | Retrospective statistics about suicide | Altitude climate | Altitude, cold and seasonal temperature changes above 25 ° C are all linked to increased suicide rates in women. |
| Burke et al., 2018 | USA & Mexico | 1968–2010 | Retrospective | General population | Not provided | Not provided | Retrospective statistics about climate | Retrospective statistics about suicide | Higher temperatures | A 1 ° C rise in monthly average temperature increases suicide rates by 0.7% in the United States and 2.1% in Mexico. |
| Fernández-Niño et al., 2018 | Colombia | 2011–2014 | Retrospective | General population | N: 1,942 ♀: 386 ♂: 1,556 | Not provided | Retrospective statistics about climate | Retrospective statistics about suicide | Air pollution | No statistically significant association between air pollution and suicide. |
| Akkaya-Kalayci et al., 2017 | Turkey | 2010 | Retrospective | Individuals with suicidal behaviour | N: 2,131 ♀: 1,740 ♂: 391 | 15–25 | Retrospective statistics about climate | Retrospective statistics about suicide | Seasonal changes & ambient temperature (sunlight duration and rainfall) | Suicide attempts in youth were associated with seasonal changes, peaking in summer, and rising temperatures, with a short-term effect observed in men after adjusting for seasonality. |

(*Continued*)

| References | Country | Study period | Study design | Population | Sample size and gender | Age | Climate measurement | Suicide measurement | Climate exposure/ climate types | Finding |
|---|---|---|---|---|---|---|---|---|---|---|
| Carleton 2017 | India | 1967–2013 | Longitudinal | General population | Not provided | Not provided | Retrospective statistics about climate | Retrospective statistics about suicide | High temperatures | High temperatures increase suicide rates, especially during the growing season, when heat reduces crop yields. An increase of 1 ° C above 20 ° C causes around 70 additional suicides per day. |
| Fernández-Arteaga et al., 2016 | Mexico | 2005–2012 | Retrospective | Individuals with suicidal behaviour | N: 1,357 ♀: 212 ♂: 1,145 | 10–76 | Retrospective statistics about climate | Retrospective statistics about suicide | Environmental temperature (rain & high temperatures) | There was an association between environmental temperature and suicide. |
| Fountoulakis et al., 2016 | Europe continent | 2000–2012 | Retrospective | General population | Not provided | Not provided | Retrospective statistics about climate | Retrospective statistics about suicide from WHO | Low temperature (cold climate) | Climate variables related to suicide risks were responsible for 37.6% for males, and 28.1% for females. |
| Fountoulakis et al., 2016 | Greece | 2000–2012 | Retrospective | General population | Not provided | Not provided | Retrospective statistics about climate | Retrospective statistics about suicide | Climate variables (high temperature) | Male suicide rates are related to minimum temperature, while female rates show no correlation. Suicide attempts peak from May to August and again from December to February. |
| Kim et al., 2016 | South Korea, Japan & Taiwan | 1972–2010 | Case-crossover | Individuals with suicidal behaviour | Not provided | Not provided | Retrospective statistics about climate | Retrospective statistics about suicide | Ambient temperature (sunshine duration & higher temperature) | There are positive associations between suicide rates and elevated ambient temperatures in three East Asian countries. |
| Bakian et al., 2015 | USA | 2000–2010 | Case-crossover | Individuals with suicidal behaviour | N: 1,546 ♀: 336 ♂: 1,210 | 35–65 | Retrospective statistics about climate | Retrospective statistics about suicide | Air Pollution | Higher levels of nitrogen dioxide and fine particulate matter (PM2.5) were linked to an increased risk of suicide, especially during seasonal transitions. The risk was higher with nitrogen dioxide in spring / fall (OR = 1.35) and fine particulate matter in spring (OR = 1.28), highlighting the impact of climate and seasonal changes. |
| Hiltunen et al., 2014 | Finland | 1974–2010 | Retrospective | General population | N: 10,802 ♀: 3,151 ♂: 7,651 | Not provided | Retrospective statistics about climate | Retrospective statistics about suicide | Ambient temperature & the timing of thermal seasons | A decrease in temperature for five days was associated with lower suicide rates in men, while the results for women were more inconsistent, suggesting a stronger link between the change in temperature and male suicide rates. |
| Helama et al., 2013 | Finland | 1751–2008 | Retrospective | General population | Not provided | Not provided | Retrospective statistics about climate | Retrospective statistics about suicide | Ambient temperature | Temperature variability accounted for more than 60% of the variance in suicide rate, with a decline in suicide rates observed despite ongoing warming after the initiation of a national prevention programme. |
| Müller et al., 2011 | Germany | 1998–2005 | Retrospective | Individuals with suicidal behaviour | N: 2,987 ♀: 1,114 ♂: 1,873 | M: 44.7 | Retrospective statistics about climate | Retrospective statistics about suicide | Solar radiation & air temperature | Higher global radiation and air temperatures on the day of and the day before suicide were associated with an increased risk of suicide. |

**Table 1.** (*Continued*)

| References | Country | Study period | Study design | Population | Sample size and gender | Age | Climate measurement | Suicide measurement | Climate exposure/ climate types | Finding |
|---|---|---|---|---|---|---|---|---|---|---|
| Lin et al., 2008 | Taiwan | 1997–2003 | Retrospective | General population | N: 18,130 | Not provided | Retrospective statistics about climate | Retrospective statistics about suicide | Seasonality & high temperature | Seasonal trends in violent suicides showed a significant peak from March to May, with rising ambient temperatures predicting higher violent suicide rates. Seasonality was not observed in non-violent suicides, suggesting that violent suicides may be more influenced by climate and biological factors. |
| Oravecz et al., 2007 | Slovenia | 1971–2002 | Retrospective | General population | N: 18,675 ♀: 4,350 ♂:14,325 | Not provided | Retrospective statistics about climate | Retrospective statistics about suicide | Seasonality & temperature amplitude | The seasonality in suicide rates decreased with time, with a slight positive correlation between the amplitude of the 12-month cycle and suicide rates, statistically significant only for women (P = 0.0053). |
| Preti et al., 2007 | Italy | 1974–2003 | Retrospective | General population | N = 97,693 ♀: 26,466 ♂: 71,227 | Not provided | Retrospective statistics about climate | Retrospective statistics about suicide | Global warming (high temperature | Rising temperatures were linked to higher suicide rates in men, particularly from May to August, with fewer suicides in January. For women, the relationship was less consistent. These results highlight the significant influence of warming temperatures on suicide rates, underscoring the potential impact of climate change. |
| Rocchi et al., 2007 | Italy | 1974–2003 | Retrospective | General population | N: 97,693 ♀: 26,466 ♂: 71,227 | Not provided | Retrospective statistics about climate | Retrospective statistics about suicide | Seasonality & temperature amplitude | Seasonality accounted for around 40% of the variance in suicide in males and 39% in females, with a peak in spring. The timing and amplitude of this peak varied over time, influenced by yearly suicide rates. |
| Deisenhammer et al., 2003 | Austria | 1995–2000 | Retrospective | Individuals with suicidal behaviour | N: 702 ♀: 184 ♂: 518 | Not provided | Retrospective statistics about climate | Retrospective statistics about suicide | High temperatures, low humidity & thunderstorm | The risk of suicide was significantly higher on days with high temperatures, low relative humidity or a thunderstorm, as well as on days following a thunderstorm. |
| Doganay et al., 2003 | Turkey | 1996–2001 | Retrospective | Individuals with suicidal behaviour | N: 1,119 ♀: 802 ♂: 317 | 15–65 | Retrospective statistics about climate | Retrospective statistics about suicide | Climatic and diurnal variation (humidity, ambient temperature, duration & intensity of sunlight) | Suicide attempts were positively correlated with humidity, temperature, sunlight, and negatively with cloudiness and atmospheric pressure. |
| Preti, 1998 | Italy | 1974–1994 | Retrospective | General population | N: 22,564 | Not provided | Retrospective statistics about climate | Retrospective statistics about suicide | Ambient temperature, humidity & sunlight | Climate factors, particularly humidity, rainfall and sunlight exposure, were significantly related to suicide rates, with higher suicide rates observed in dry areas with less sunlight exposure. |

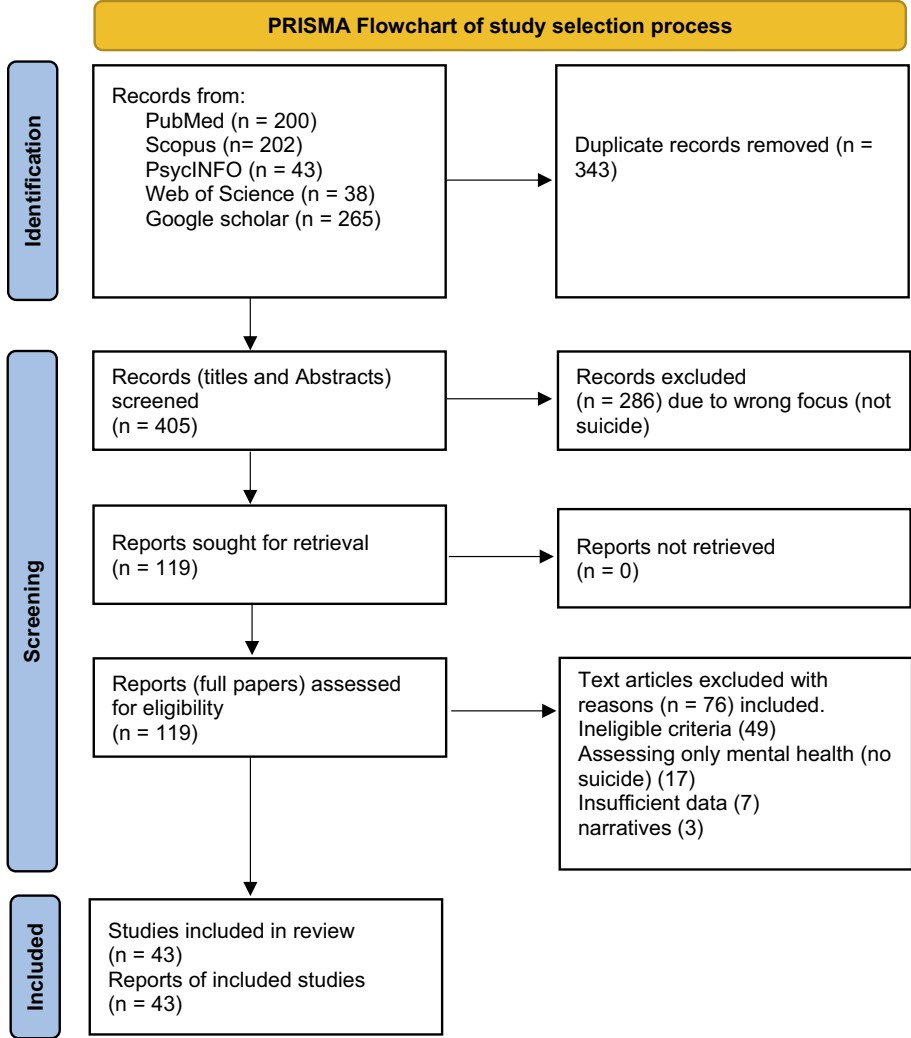

**Figure 1.** PRISMA flowchart of the study selection.

widely, ranging from small cohorts of a few hundred participants to large-scale studies involving tens of thousands. Although some included more than 20,000 participants (Preti, 1997; Preti et al., 2007; Rocchi et al., 2007; Asirdizer et al., 2018; Lee et al., 2019; Bozsonyi et al., 2020; Rahman et al., 2023; Villeneuve et al., 2023), the majority were limited to only 2000 (Deisenhammer et al., 2003; Doganay et al., 2003; Müller et al., 2011; Bakian et al., 2015a; Fernández-Arteaga et al., 2016; Akkaya-Kalayci et al., 2017; Fernández-Niño et al., 2018; Matthews et al., 2019; Koszewska et al., 2019; Di Nicola et al., 2020; Makris et al., 2021; Aguglia et al., 2021). Most subjects were between 35 and 65 years of age with only a few exceptions in which populations of under-aged (15–25) (Akkaya-Kalayci et al., 2017) or over-aged (Lee et al., 2019; Koszewska et al., 2019; Schneider et al., 2020) populations were exclusively targeted. Females outnumbered males in some studies (Doganay et al., 2003; Akkaya-Kalayci et al., 2017; Matthews et al., 2019; Di Nicola et al., 2020; Bauer et al., 2021a; Aguglia et al., 2021), leaving male predominance in the majority of the remaining studies (Deisenhammer et al., 2003; Oravecz et al., 2007; Preti et al., 2007; Müller et al., 2011; Hiltunen et al., 2014; Bakian et al., 2015a; Fernández-Arteaga et al., 2016; Asirdizer et al., 2018; Fernández-Niño et al., 2018; Koszewska et al., 2019; Lee et al., 2019;

Bozsonyi et al., 2020; Kayipmaz et al., 2020; Schneider et al., 2020; Makris et al., 2021; Edwards et al., 2023). Table 1.

### Measurement of climate exposure and suicidality

The included studies employed diverse methods to measure both climate exposures and suicide outcomes, with most relying on retrospective statistical records. Climate data were largely based on core meteorological variables such as temperature anomalies, seasonal changes, precipitation, humidity, wind speed and daily sunshine duration. Several studies specifically examined air pollution (e.g., PM2.5, $NO_2$, $O_3$, $SO_2$) and its association with suicidality, while others assessed solar insolation and extreme weather events such as heatwaves, floods and droughts. A smaller number used specialized instruments, including the Climate Change Stress and Impairment Scale (CC-DIS) and the METEO-Q, to capture climate-related distress and sensitivity. Suicide outcomes were primarily measured using medical records, national suicide registries and hospital databases, often aligned with classifications from the Diagnostic and Statistical Manual of Mental Disorders (DSM). In addition, several studies employed validated psychometric scales, such as the Suicide Ideation and Behavior Scale (SIBS) and the Implicit Self-Harm Association Test (IAT), to assess suicidal

ideation and related behaviours. Reported outcomes included suicidal thoughts, attempts, self-harm, severity of attempts and distinctions between violent and non-violent methods. Some studies also focused on specific populations (e.g., students, employees, pregnant individuals or suicide attempters) to examine subgroup differences in climate-related suicide risk. Overall, most studies combined quantitative climate data with registry-based or self-reported measures of suicidality to explore this relationship (see Table 1 for details).

### The outcome of risk of bias assessments

Quality assessment using MMAT (Nha HONG et al., n.d.) of 17 studies (Bakian et al., 2015a; Burke et al., 2018; Matthews et al., 2019; Lee et al., 2019; Kayipmaz et al., 2020; Di Nicola et al., 2020; Makris et al., 2021; Bergmans et al., 2021; Jenwitheesuk et al., 2022; Miyazaki et al., 2023; Rahman et al., 2023; Villeneuve et al., 2023; Brailovskaia and Teismann, 2024; Hertzog et al., 2024; Lian et al., 2024; Ndetei et al., 2024; Runkle et al., 2024) showed that most of the studies had clear research questions and used data well suited to address those questions. Sample strategies were generally appropriate for the research objectives and, in most cases, the samples appeared to represent the target population. The measurements used in the studies were relevant and appropriate, and the statistical analyses were suitable to answer the research questions. On the contrary, when it came to assessing the risk of non-response bias, none of the studies provided enough information to determine this, resulting in all studies being marked as 'Cannot tell' for this criterion. The outcome of the MMAT quality assessment is shown in Table 2.

The quality assessment using ROBINS-E (Higgins et al., 2024) of 26 studies (Preti, 1997; Deisenhammer et al., 2003; Doganay et al., 2003; Oravecz et al., 2007; Preti et al., 2007; Rocchi et al., 2007; Lin et al., 2008; Müller et al., 2011; Helama et al., 2013; Hiltunen et al., 2014; Kim et al., 2016; Fernández-Arteaga et al., 2016a; Fountoulakis, Chatzikosta, et al., 2016; Fountoulakis, Savopoulos, et al., 2016a; Akkaya-Kalayci et al., 2017; Carleton, 2017; Asirdizer et al., 2018; Fernández-Niño et al., 2018; Koszewska et al., 2019; Bozsonyi et al., 2020; Schneider et al., 2020; Aguglia et al., 2021; Bauer et al., 2021a; Cheng et al., 2021; Freichel & O'Shea, 2023; Edwards et al., 2023) showed that all the studies had 'Some concerns' in Domain 1, which relates to bias due to confounding. Domains 2 and 6, covering bias arising from exposure measurement and bias arising from measurement of the outcome, were generally rated 'low' risk. Domain 3, which examines bias in the selection of participants in the study, and Domain 5, which addresses bias due to missing data, showed mixed ratings, with several studies indicating 'Some concerns'. Domain 4, which addressed bias due to post-exposure interventions, had insufficient information. Overall, the studies predominantly fell under the category 'Some concerns', indicating a moderate risk of bias in general. The outcome of the ROBINS-E quality assessment is illustrated in Figures 2 and 3.

### Main findings

#### Distribution of climate indictors and suicidality outcomes

To illustrate how climate indicators and suicidality outcomes align across literature, Table 3 summarizes the distribution of the 43 included studies. The majority examined temperature or heat as the main indicator (28 studies), followed by air pollution (5 studies), natural disasters such as floods or droughts (2 studies), and seasonality or solar-related variables (2 studies). Six studies focused on other indicators, such as eco-anxiety, altitude, foehn wind or aeroallergens. Regarding suicidality outcomes, 41 studies investigated fatal suicide, while only 2 assessed suicidal ideation, reflecting the spectrum of suicide outcomes from thoughts to fatal acts. These two suicidal ideation studies, conducted among students and adolescents, reported significant associations between climate-related stressors and suicidal thoughts but were limited to specific populations. This imbalance highlights the dominance of mortality-focused research and the lack of attention to earlier stages of suicidality.

#### Geographic distribution and climate factors in suicide risk

Although LAMICs cover a large portion of the global population, there are relatively few studies that examine the connection between climate change and suicidality in these regions. Most of the research on suicidality linked to climate change has been conducted in European countries (Aguglia et al., 2021; Akkaya-Kalayci et al., 2017a; Asirdizer et al., 2018; Bozsonyi et al., 2020; Brailovskaia and Teismann, 2024; Di Nicola et al., 2020a; Doganay et al., 2003; Helama et al., 2013; Kayipmaz et al., 2020; Müller et al., 2011; Oravecz et al., 2007; Preti, 1997, 1998a; Preti et al., 2007; Schneider et al., 2020), USA (Bergmans et al., 2021; Rahman et al., 2023; Freichel & O'Shea, 2023; Runkle et al., 2024), Australia (Matthews et al., 2019; Edwards et al., 2023; Hertzog et al., 2024) and Canada (Villeneuve et al., 2023; Freichel & O'Shea, 2023). The sociocultural and environmental contexts, as well as resource availability in LAMICs, differ significantly from those of developed European nations. Therefore, there is a pressing need for more regional research to develop remediation strategies specific to these areas.

#### Temperature and heat exposure and suicide

Temperature and heat anomalies were among the most strongly linked climate factors to suicidality. Multiple studies found that higher temperatures significantly increased suicide mortality, particularly in older adults and men (Preti et al., 2007; Cheng et al., 2021; Hertzog et al., 2024). In Australia, 55-year-old men and older showed the highest vulnerability to heat-related suicides, while studies in the United States, Canada and Germany reported that each 1 °C increase in temperature was associated with a rise in suicide rates (Burke et al., 2018; Schneider et al., 2020; Cheng et al., 2021; Hertzog et al., 2024). Furthermore, Bozsonyi et al. (2020)) in Hungary found a strong positive correlation between suicide rates, high ambient temperatures and duration of sunshine. Kayipmaz et al. (2020)) in Turkey reported that an increase in minimum temperature on the day of application was correlated with a rise in suicides. Similarly, in Mexico, rain and high temperatures were significantly associated with suicide risk in which a 1 °C increase in monthly temperature led to a 2.1% increase in suicide rates, while in the United States, the increase was 0.7% (Fernández-Arteaga et al., 2016; Burke et al., 2018).

In India, high temperatures increased suicides, particularly during the growing season, suggesting that economic hardship due to climate-related crop failures exacerbated the risk of suicide (Carleton, 2017). In particular, older adults (65+) in Sweden showed increased vulnerability to suicidal risk due to higher temperatures and increased sunshine, although these effects were not significant after adjustments (Makris et al., 2021). Furthermore Fountoulakis, Chatzikosta, et al. (2016) and Fountoulakis, Savopoulos, et al. (2016b) found that male suicide rates in Greece and throughout Europe were linked to minimum temperatures, with suicide attempts peaking during May to August and December to February. Particularly low temperatures accounted for 37.6% of

**Table 2.** Outcome of quality assessment using the Mixed Methods Appraisal Tool (MMAT)

| First author | Screening questions | | Quantitative descriptive studies | | | | |
|---|---|---|---|---|---|---|---|
| | 1. Are there clear research questions? | 2. Do the collected data allow to address the research questions? | 3. Is the sampling strategy relevant to address the research question? | 4. Is the sample representative of the target population? | 5. Are the measurements appropriate? | 6. Is the risk of non-response bias low? | 7. Is the statistical analysis appropriate to answer the research question? |
| Brailovskaia and Teismann, 2024 | Yes | Yes | Yes | No | Yes | Cannot tell | Yes |
| Hertzog et al., 2024 | Yes | Yes | Yes | Yes | Yes | Cannot tell | Yes |
| Lian et al., 2024 | Yes | Yes | Yes | Yes | Yes | Cannot tell | Yes |
| Ndetei et al., 2024 | Yes | Yes | Yes | Yes | Yes | Cannot tell | Yes |
| Runkle et al., 2024 | Yes | Yes | Yes | Cannot tell | Yes | Cannot tell | Yes |
| Miyazaki et al., 2023 | Yes | Yes | Yes | Yes | Yes | Cannot tell | Yes |
| Rahman et al., 2023 | Yes | Yes | Yes | Cannot tell | Yes | Cannot tell | Yes |
| Villeneuve et al., 2023 | Yes | Yes | Yes | Yes | Yes | Cannot tell | Yes |
| Jenwitheesuk et al., 2022 | Yes | Yes | Yes | Yes | Yes | Cannot tell | Yes |
| Bergmans et al., 2021 | Yes | Yes | Yes | Cannot tell | Yes | Cannot tell | Yes |
| Makris et al., 2021 | Yes | Yes | Yes | Yes | Yes | Cannot tell | Yes |
| Kayipmaz et al., 2020 | Yes | Yes | Yes | Cannot tell | Yes | Cannot tell | Yes |
| Di Nicola et al., 2020 | Yes | Yes | Yes | Yes | Yes | Cannot tell | Yes |
| Lee et al., 2019 | Yes | Yes | Yes | Yes | Yes | Cannot tell | Yes |
| Matthews et al., 2019 | Yes | Yes | Yes | No | Yes | Cannot tell | Yes |
| Burke et al., 2018 | Yes | Yes | Yes | Yes | Yes | Yes | Yes |
| Bakian et al., 2015 | Yes | Yes | Yes | Cannot tell | Yes | Cannot tell | Yes |

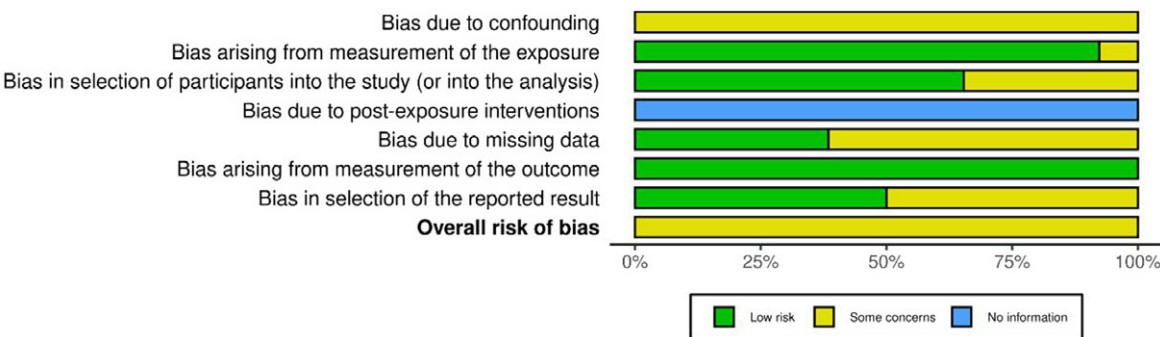

**Figure 2.** ROBINS-E assessment of individual studies across domains. Most studies showed 'Some concerns' in Domain 1 (bias due to confounding). Domains 2 and 6 (bias in exposure measurement and outcome measurement) were generally rated 'low' risk. Domains 3 (bias in selection of participants) and 5 (bias due to missing data) showed mixed ratings, while Domain 4 (bias due to post-exposure interventions) often lacked sufficient information.

male suicides and 28.1% of female suicides. Furthermore, the timing of heat exposure was critical. Studies in Germany, Italy and Austria found that high temperatures, low humidity and thunderstorms were significantly linked to higher suicide rates, men being more affected than women (Preti, 1998; Deisenhammer et al., 2003; Müller et al., 2011).

### Air pollution and suicide
Exposure to air pollution was also a key factor contributing to suicidality. In Japan, elevated SO2 levels were strongly linked to higher suicide attempts, while NO (nitrogen oxide) levels showed an inverse relationship with suicidality (Miyazaki et al., 2023). In Thailand, a 1 mg/m$^3$ increase in PM2.5 dust led to a 63.5% increase in suicide risk, demonstrating the direct impact of poor air quality (Jenwitheesuk et al., 2022). Studies in Canada and the United States showed that exposure to fine particulate matter (PM2.5) and nitrogen dioxide (NO2) significantly increased suicide risks, particularly during seasonal transitions in spring and fall (Bakian et al., 2015b; Rahman et al., 2023; Villeneuve et al., 2023). Furthermore, Lian et al. (2024)) in China found that short-term exposure to air pollution significantly increased suicide deaths, particularly in cities at the county level, accounting for up to 8.07% of excess suicide deaths. Similarly, in Italy, solar radiation (65%) and PM2.5 (32%) were positively correlated with suicide attempts, particularly among middle-aged males (Aguglia et al., 2021). While many studies confirmed the association between air pollution and suicide, some did not find significant links. In Colombia, no statistically significant association was found between air pollution and suicidality, suggesting potential geographical and demographic differences in susceptibility (Fernández-Niño et al., 2018).

### Seasonality and suicide
Seasonal variations in temperature and exposure to sunlight were repeatedly linked to suicide rates. In studies from the United States, United Kingdom, Canada, Taiwan and Italy, suicidal ideation and attempts peaked during seasonal transitions, with spring and early summer showing the highest suicide rates (Preti, 1997; Lin et al., 2008; Freichel & O'Shea, 2023c). Suicide rates were observed to peak in December, and implicit self-harm thoughts reaching their highest levels in February, suggesting a delayed psychological effect of reduced daylight exposure (Freichel & O'Shea, 2023c). Kim et al. (2015) in South Korea, sJapan and Taiwan found a positive association between suicide rates and elevated ambient temperatures, including the duration of sunshine. Akkaya-Kalayci et al. (2017) in

Turkey found that suicide attempts in youth were linked to seasonal changes, peaking in summer with short-term effects in men. Koszewska et al. (2019) in Poland found that the foehn wind was associated with increased suicide risks, particularly in summer and autumn. Furthermore, in Slovenia, the seasonality of suicide showed a stronger correlation among women (Oravecz et al., 2007). In Italy, suicide attempts peaked in spring, accounted for 40% in males and 39% in females, while in Taiwan, rising ambient temperatures from March to May were linked to higher violent suicide rates (Rocchi et al., 2007; Lin et al., 2008). Furthermore, Helama et al. (2013)) in Finland found that temperature variability accounted for more than 60% of the variance in suicide rates. However, a decline in suicide rates was observed despite ongoing warming.

### Natural disasters and extreme climate events and suicide
Several studies linked natural disasters, such as wildfires, floods and droughts, to increased suicidal thoughts and behaviours. In Australia, floods were associated with a 7% increase in suicidal ideation, particularly among adolescents (Matthews et al., 2019; Edwards et al., 2023). Lee et al. (2019) in South Korea found that dust storms were associated with a higher risk of suicide, with stronger associations during prolonged storms. A study in Turkey found that altitude-related climate conditions contributed to higher suicide rates, particularly among women (Asirdizer et al., 2018). Studies in Turkey and Italy further emphasized the role of humidity, thunderstorms and extreme weather in increasing the risk of suicide (Doganay et al., 2003; Di Nicola et al., 2020). Hiltunen et al. (2014) in Finland found that a decrease in temperature over five days was associated with lower male suicide rates, suggesting stronger climate–suicide interactions. Additionally, a study in the United States found that grass pollen exposure was linked to suicide on the same day, particularly among women and individuals with lower education levels (Bergmans et al., 2021).

### Climate change anxiety and suicide
Several studies highlighted the impact of climate change distress on suicidality. Psychological distress linked to climate change-related stressors was found to be a significant risk factor for suicidal ideation among university students and employees in Germany (Brailovskaia and Teismann, 2024). Similarly, among high school students in Kenya, perceived threats of climate change were associated with an increased risk of suicidal thoughts and attempts (Ndetei et al., 2024).

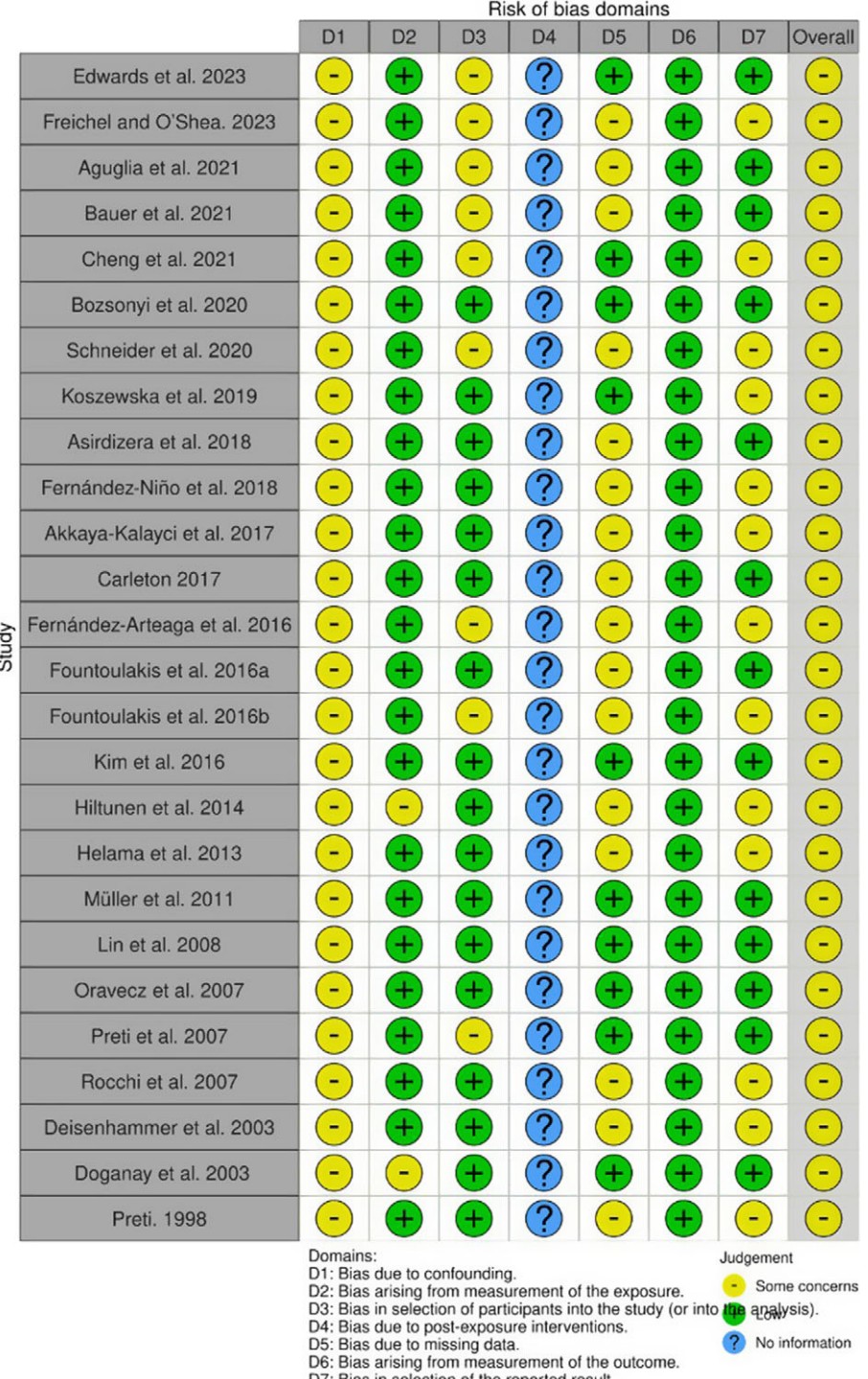

**Figure 3.** Summary of ROBINS-E quality assessment across all included studies. The majority of studies fell under the category of 'Some concerns', reflecting a moderate overall risk of bias.

### Vulnerable populations

Vulnerable populations, including older adults, pregnant women, migrant populations and people with mental illness, exhibit a greater susceptibility to climate-related suicide risks. Studies in individuals with bipolar disorder indicate that increased sensitivity to weather changes is correlated with a higher risk of suicide attempts (Di Nicola et al., 2020). Research in 40 countries found that significant seasonal changes in solar insolation increased the likelihood of suicidal ideation and attempts in individuals with bipolar disorder (Bauer et al., 2021b). Pregnant women also face increased vulnerability, with a study reporting a 28% increase in suicidal thoughts for every 5 °C temperature increase (Runkle et al., 2024). Similarly, older adults (≥65 years) experience greater sensitivity to high temperatures and seasonal variations, with stronger effects observed during summer and autumn (Schneider et al., 2020).

**Table 3.** Distribution of climate indicators and suicidality outcomes across included studies (n = 43)

| Climate indictors | Suicidality | | Total studies |
|---|---|---|---|
| | Ideation | Fatal suicide | |
| Temperature / Heat | 0 | 28 | 28 |
| Air pollution | 1 | 4 | 5 |
| Natural disasters | 1 | 1 | 2 |
| Seasonality / Sunlight | 0 | 2 | 2 |
| Other (eco-anxiety, etc.) | 0 | 6 | 6 |
| Total | 2 | 41 | 43 |

## Discussion

This systematic review investigated the complex relationship between climate change and suicidality, synthesizing findings from 43 studies conducted throughout the world. The results highlighted the link between climate indicators, such as rising temperatures, air pollution and extreme weather events, and suicidality, with important implications for public health, particularly among vulnerable populations.

Most of the research related to climate change and suicidality comes from high-income countries, while LAMICs, which are disproportionately exposed to extreme climate events, remain severely underrepresented in the existing evidence base. Although the 43 included studies represent 13 countries worldwide, the overwhelming majority originate from Europe, North America and Australia. This imbalance represents a major structural limitation of the field, rather than this review alone, given that many of the world's most climate-vulnerable populations reside in LAMICs where climate shocks, food insecurity, forced displacement, political instability and poverty are more prevalent. In such settings, climate stressors may interact more strongly with adverse social determinants of health and limited access to mental health services, potentially amplifying suicide risk beyond what is observed in high-income countries, while protective community factors remain poorly understood. The absence of robust suicidality data from these regions is therefore a critical global evidence gap that limits equitable prevention planning and policy development. People exposed to extreme weather, high temperatures, rainfall, air pollutants, floods and humidity demonstrate high suicidal behaviour (from ideation to fatal outcomes). A positive association between seasonal variation and suicidal behaviour is also reported, particularly in women. While two studies provided important evidence on climate-related suicidal ideation, their limited number and population-specific focus underscore the critical lack of research on non-fatal stages of suicidality compared with the extensive focus on fatal suicide. The review revealed a consistent association between increased ambient temperature and increased risk of fatal suicide. For example, studies in the United States, Mexico and India demonstrated a clear increase in suicide rates correlated with elevated temperatures and prolonged heatwaves. This is consistent with physiological and psychological mechanisms in which high temperatures exacerbate psychological problems such as aggression, irritability and impulsivity, which can lead to increased suicidal behaviour (Hou et al., 2023; Fischer et al., 2024). Evidence from meta-analyses also supported that increased temperature and temperature variability could be associated with an increase in suicide and suicidal behaviour (Björkstén et al., 2005; Thompson et al., 2023; To et al., 2024). Additionally, seasonal patterns observed in several studies indicated higher rates of suicidality during the summer months, particularly among women, suggesting the possibility of female gender in climatic adversities. Climatic changes and an extreme rise in temperature have been reported in recent decades globally. This trend is expected to continue in the coming days and is likely to influence suicidal behaviour. Hence, it is important to consider climatic adversities in the suicide-prevention programme.

Extreme weather events, such as floods, hurricanes and droughts, were also linked to adverse mental health outcomes, including PTSD, depression and suicidality (Cianconi et al., 2020; Walinski et al., 2023; Heanoy and Brown, 2024; Patwary et al., 2024). These findings highlight the psychosocial burden of climate disasters, driven by loss of livelihoods, displacement and community disintegration. Such disasters are common in specific geographical regions (more so in coastal areas). Unfortunately, the disaster-affected countries have little research on suicidal behaviour in reference to the disasters. Therefore, it is important to have more research in these countries on the mental health aspects of disasters to understand the region-specific needs.

Furthermore, the review identified air pollution as a compounding factor, with pollutants such as particulate matter and nitrogen dioxide contributing to poor mental health and suicide risk (Braithwaite et al., 2019; Bhui et al., 2023; Nobile et al., 2023). Measures being taken to control pollution should aim at reducing the level of nitrogen dioxide (decreasing its production by intervening at the levels of sources from where it is being produced) and particulate matter suspended in air.

### Methodological considerations

Importantly, all studies assessed using the ROBINS-E tool were judged to have some concerns regarding confounding bias, particularly due to unmeasured or residual factors such as socioeconomic conditions, unemployment, substance use, comorbid mental disorders, social isolation and access to mental health services. This pervasive limitation reflects the inherent challenges of observational climate–health research rather than methodological weaknesses of individual studies. Consequently, the associations identified in this review should not be interpreted as causal effects but rather as population-level correlations between climate indicators and suicidality that may be shaped by broader contextual and social determinants. This consideration warrants cautious interpretation of effect estimates and highlights the need for future longitudinal and quasi-experimental research designs that can better address confounding. The prevalence of cross-sectional and retrospective study designs restricts causal inferences, highlighting a methodological gap in the literature. Longitudinal studies are essential to elucidate the temporal dynamics between climate change and suicidality. Furthermore, the variability in climate and suicidality measures in all studies complicates comparability and synthesis. Standardizing methodologies, including consistent definitions and measurement tools for climate indicators and suicidal behaviours, could enhance the robustness of future research. The methods used, the parameters studied and the results measured in most existing research are heterogeneous; therefore, it is challenging to accurately compare the outcomes between the studies.

### Recommendations and implications for policy and practice

The findings of this review have direct and actionable implications for public health policy and suicide prevention practice. *First,*

climate-sensitive mental health early warning systems should be developed in regions experiencing recurrent heatwaves and extreme weather events. Several included studies demonstrated significant increases in suicide risk at temperature rises of 1–5 °C, suggesting that locally calibrated temperature thresholds could serve as triggers for activating mental health surge responses, crisis hotlines and outreach services to vulnerable populations such as older adults, individuals with mental illness, pregnant women and outdoor workers. *Second*, establishing publicly accessible cooling centres in different urban and rural locations during periods of extreme summer heat represents a feasible and cost-effective preventive intervention. These centres can support both physical and mental health, reduce heat-related psychological distress, stabilize autonomic nervous system responses and potentially mitigate heat-associated risks of suicidal behaviour, particularly among vulnerable populations who lack access to adequate cooling at home. *Third*, air-quality monitoring systems should be integrated with suicide surveillance, particularly in urban and industrial areas with high concentrations of $PM_{2.5}$, $NO_2$ and $SO_2$. Public health agencies should issue combined air-pollution and mental-health alerts, alongside temporary restrictions on outdoor work and targeted psychosocial support during peak pollution periods. *Fourth*, standardized climate-suicidality measurement protocols are urgently needed to improve comparability across studies and surveillance systems. These should include harmonized definitions of heat exposure, extreme events and suicidality outcomes, as well as routine collection of data on suicidal ideation, attempts and fatal suicide, not mortality alone. *Fifth*, disaster preparedness and response plans should formally include mental health and psychosocial support (MHPSS), including digital and electronic mental health (Ahmed and Heun, 2024) components tailored to climate-related disasters such as floods, droughts, dust storms and wildfires, particularly to ensure continuity of care in remote and hard-to-access areas during disasters. This should involve pre-positioned psychological first aid teams, mobile mental health units and structured community follow-up for affected populations. *Finally*, in LAMICs, implementation of these interventions must account for resource constraints, limited mental health workforces and fragile health systems. Climate–mental health early warning systems may therefore be most feasible when integrated into existing meteorological, primary healthcare and civil protection infrastructures rather than developed as stand-alone platforms. Cooling centres can be adapted using low-cost community spaces such as schools, mosques, churches or municipal buildings, supported by local governments, NGOs and community volunteers. Similarly, eMHPSS approaches offer particular promise in LAMICs by enabling task-shifting, remote psychological first aid and continuity of care during climate disasters, especially where specialist services are scarce. Leveraging mobile phone penetration, community health workers and partnerships with humanitarian organizations may allow scalable and cost-effective implementation in resource-limited settings.

## Strengths and limitations

The strengths of this review stem from being the first systematic review on this topic, providing a comprehensive analysis of 43 studies in various geographic regions and climate indicators, offering solid information on the relationship between climate change and suicidality. Furthermore, the review identifies notable research gaps, offering actionable recommendations for future studies and filling a critical public health evidence gap. However, limitations persist, including the predominance of cross-sectional and retrospective studies, which restrict causal inferences, and the exclusion of qualitative research may overlook valuable insights into lived experiences and cultural contexts of climate change and suicidality. Additionally, the reliance on secondary data limits the ability to assess study quality and methodological rigor directly, also the heterogeneity in study designs and methodologies complicates the synthesis of findings, and the underrepresentation of LAMICs limits the global applicability of the conclusions. Furthermore, the restriction to English-language publications may have led to the exclusion of relevant studies conducted especially in climate-vulnerable LAMIC regions and published in local languages; future multilingual systematic reviews may therefore help to partially address the geographic and socioeconomic evidence gaps identified in this review. Although qualitative studies were excluded due to the review's focus on quantitatively measurable associations, this represents an important conceptual limitation. Qualitative research can provide valuable insights into lived experiences of climate-related distress, cultural interpretations of environmental stressors, help-seeking behaviours and barriers to mental health care. Such insights are particularly relevant for understanding how and why climate exposures translate into suicidality in specific contexts and can meaningfully inform the design, acceptability and cultural appropriateness of preventive interventions. Future reviews integrating mixed-methods evidence may therefore strengthen intervention development and implementation strategies.

## Conclusion

The relationship between climate change and suicidality represents an emerging public health crisis with extensive implications. Evidence indicated that higher temperatures and extreme weather events increase the risks of suicide. By promoting research, improving policy integration and strengthening community resilience, it can alleviate the mental health consequences of climate change and build a more sustainable and equitable future. This undertaking requires collective action, connecting disciplines, regions and sectors to protect mental health in the context of a changing climate. Climate is a dynamic state that is constantly changing. In the coming days we are expected to see significant global climate change, likely significantly impacting mental well-being. Therefore, it is important to keep track of suicidal behaviour in relation to climate change around the world.

**Open peer review.** To view the open peer review materials for this article, please visit http://doi.org/10.1017/gmh.2026.10176.

**Data availability statement.** All data generated or analysed during this study are included in this published article. The full search strategy and extracted dataset are available from the corresponding author upon reasonable request.

**Author contributions.** D.R.A. conceived and initiated the study idea and led the study design, contextualization, illustration, data interpretation, study selection, literature search, data extraction, manuscript writing and critical revision. S.K.K. contributed to manuscript writing and revision. M.D.A. contributed to the literature search, study selection, quality assessment and risk of bias assessment. R.H. provided critical feedback and substantive comments on the manuscript. All authors reviewed and approved the final version of the manuscript.

**Financial support.** This study received no specific grant from any funding agency in the public, commercial or not-for-profit sectors and was conducted solely using the authors' own time and resources.

**Competing interests.** The authors declare that they have no competing interests.

**Ethical approval.** Ethical approval was not required for this study as it is a systematic review based exclusively on previously published data and did not involve direct contact with human participants. The review protocol was prospectively registered in PROSPERO under the registration number CRD42024539305.

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

## Appendix

| Section and topic | Item # | Checklist item | Location where item is reported |
|---|---|---|---|
| **TITLE** | | | |
| Title | 1 | Identify the report as a systematic review. | |
| **ABSTRACT** | | | |
| Abstract | 2 | See the PRISMA 2020 for Abstracts checklist. | |
| **INTRODUCTION** | | | |
| Rationale | 3 | Describe the rationale for the review in the context of existing knowledge. | |
| Objectives | 4 | Provide an explicit statement of the objective(s) or question(s) the review addresses. | |
| **METHODS** | | | |
| Eligibility criteria | 5 | Specify the inclusion and exclusion criteria for the review and how studies were grouped for the syntheses. | |
| Information sources | 6 | Specify all databases, registers, websites, organizations, reference lists and other sources searched or consulted to identify studies. Specify the date when each source was last searched or consulted. | |
| Search strategy | 7 | Present the full search strategies for all databases, registers and websites, including any filters and limits used. | |
| Selection process | 8 | Specify the methods used to decide whether a study met the inclusion criteria of the review, including how many reviewers screened each record and each report retrieved, whether they worked independently, and if applicable, details of automation tools used in the process. | |
| Data collection process | 9 | Specify the methods used to collect data from reports, including how many reviewers collected data from each report, whether they worked independently, any processes for obtaining or confirming data from study investigators, and if applicable, details of automation tools used in the process. | |
| Data items | 10a | List and define all outcomes for which data were sought. Specify whether all results that were compatible with each outcome domain in each study were sought (e.g. for all measures, time points, analyses), and if not, the methods used to decide which results to collect. | |
| | 10b | List and define all other variables for which data were sought (e.g. participant and intervention characteristics, funding sources). Describe any assumptions made about any missing or unclear information. | |
| Study risk of bias assessment | 11 | Specify the methods used to assess risk of bias in the included studies, including details of the tool(s) used, how many reviewers assessed each study and whether they worked independently, and if applicable, details of automation tools used in the process. | |
| Effect measures | 12 | Specify for each outcome the effect measure(s) (e.g. risk ratio, mean difference) used in the synthesis or presentation of results. | |
| Synthesis methods | 13a | Describe the processes used to decide which studies were eligible for each synthesis (e.g. tabulating the study intervention characteristics and comparing against the planned groups for each synthesis (item #5)). | |
| | 13b | Describe any methods required to prepare the data for presentation or synthesis, such as handling of missing summary statistics, or data conversions. | |
| | 13c | Describe any methods used to tabulate or visually display results of individual studies and syntheses. | |
| | 13d | Describe any methods used to synthesize results and provide a rationale for the choice(s). If meta-analysis was performed, describe the model(s), method(s) to identify the presence and extent of statistical heterogeneity, and software package(s) used. | |
| | 13e | Describe any methods used to explore possible causes of heterogeneity among study results (e.g. subgroup analysis, meta-regression). | |
| | 13f | Describe any sensitivity analyses conducted to assess robustness of the synthesized results. | |
| Reporting bias assessment | 14 | Describe any methods used to assess risk of bias due to missing results in a synthesis (arising from reporting biases). | |
| Certainty assessment | 15 | Describe any methods used to assess certainty (or confidence) in the body of evidence for an outcome. | |

(*Continued*)

| Section and topic | Item # | Checklist item | Location where item is reported |
|---|---|---|---|
| **RESULTS** | | | |
| Study selection | 16a | Describe the results of the search and selection process, from the number of records identified in the search to the number of studies included in the review, ideally using a flow diagram. | |
| | 16b | Cite studies that might appear to meet the inclusion criteria, but which were excluded, and explain why they were excluded. | |
| Study characteristics | 17 | Cite each included study and present its characteristics. | |
| Risk of bias in studies | 18 | Present assessments of risk of bias for each included study. | |
| Results of individual studies | 19 | For all outcomes, present, for each study: (a) summary statistics for each group (where appropriate) and (b) an effect estimate and its precision (e.g. confidence/credible interval), ideally using structured tables or plots. | |
| Results of syntheses | 20a | For each synthesis, briefly summarise the characteristics and risk of bias among contributing studies. | |
| | 20b | Present results of all statistical syntheses conducted. If meta-analysis was done, present for each the summary estimate and its precision (e.g. confidence/credible interval) and measures of statistical heterogeneity. If comparing groups, describe the direction of the effect. | |
| | 20c | Present results of all investigations of possible causes of heterogeneity among study results. | |
| | 20d | Present results of all sensitivity analyses conducted to assess the robustness of the synthesized results. | |
| Reporting biases | 21 | Present assessments of risk of bias due to missing results (arising from reporting biases) for each synthesis assessed. | |
| Certainty of evidence | 22 | Present assessments of certainty (or confidence) in the body of evidence for each outcome assessed. | |
| **DISCUSSION** | | | |
| Discussion | 23a | Provide a general interpretation of the results in the context of other evidence. | |
| | 23b | Discuss any limitations of the evidence included in the review. | |
| | 23c | Discuss any limitations of the review processes used. | |
| | 23d | Discuss implications of the results for practice, policy, and future research. | |
| **OTHER INFORMATION** | | | |
| Registration and protocol | 24a | Provide registration information for the review, including register name and registration number, or state that the review was not registered. | |
| | 24b | Indicate where the review protocol can be accessed, or state that a protocol was not prepared. | |
| | 24c | Describe and explain any amendments to information provided at registration or in the protocol. | |
| Support | 25 | Describe sources of financial or non-financial support for the review, and the role of the funders or sponsors in the review. | |
| Competing interests | 26 | Declare any competing interests of review authors. | |
| Availability of data, code and other materials | 27 | Report which of the following are publicly available and where they can be found: template data collection forms; data extracted from included studies; data used for all analyses; analytic code; any other materials used in the review. | |

