## [Reviewer Report]

This systematic review presents a comprehensively conducted examination of the association between climate change indicators and suicidality. The authors effectively establish the research rationale by positioning climate change as an emerging public health crisis with profound mental health implications, particularly highlighting the gap in research from low- and middle-income countries where climate impacts may be most severe. The introduction appropriately contextualises the review within WHO’s global research priorities for climate change and health while clearly articulating the need to synthesise fragmented evidence linking climate indicators such as temperature, air pollution, extreme weather, and seasonality to suicidality outcomes. Methodologically, the study demonstrates comprehensiveness through PRISMA-compliant systematic review procedures, comprehensive database searching, transparent study selection using Rayyan AI with dual independent screening and third-party resolution of disagreements, and appropriate dual quality assessment using both MMAT for diverse study designs and ROBINS-E for observational exposure studies. The decision to conduct narrative synthesis rather than meta-analysis is appropriately justified given substantial heterogeneity in study designs, measurement approaches, populations, and climate exposures. Minor revisions needed include: (1) clarifying the rationale for excluding qualitative studies beyond lack of effect size estimates, as the authors acknowledge this exclusion “may overlook valuable insights into lived experiences and cultural contexts” yet do not explain whether mixed-methods studies with quantitative components were considered, (2) addressing an apparent inconsistency where Table 3 reports 2 studies examining suicidal ideation yet the abstract and discussion emphasise the dramatic under-representation of non-fatal suicidality research without highlighting these two studies' contributions, and (3) strengthening the policy implications section by providing more concrete, actionable recommendations beyond general statements about “integrating climate considerations into mental health policies”. For example, specifying threshold temperatures for mental health early warning systems or proposing standardized climate-suicidality measurement protocols for future research.

---

## [Editor Report]

Dear Prof Ahmed,

Thank you for your submission. This is a timely and important topic, and as the reviewers indicate, it is a methologically sound and informative review.

My largest concern – which has partially been addressed by the authors – is the absence of LAMIC data in the review. In my mind, the absence of LAMIC data, where most extreme climate events are likely to occur (please correct me if I am wrong), and where the effects of these climate events are likely to be felt most strongly by the population, is a major flaw in the field (not this review specifically). I encourage the authors to more strongly highlight the absence of data from LAMICs. 

Furthermore, please discuss if the findings are likely to differ when considering LAMIC countries, given the difference in social determinants; and whether a future review with non-english literature is likely to address some of these gaps.

I also have one minor comment:

In the discussion, there is a statement saying “being the first study on this topic”, please change to “review”

Thank you and all the best,

Dr Sandersan Onie

---

## [Reviewer Report]

This systematic review examines climate change and suicidality across 43 studies, filling a critical public health evidence gap. The methodology is strong, PRISMA-compliant, comprehensive five-database search screening 748 articles, dual independent review, and quality assessment via MMAT and ROBINS-E.

Key findings show clear climate-suicide associations: each 1°C temperature increase correlates with, higher suicide rates; air pollutants, significantly increase suicide risk; seasonal patterns peak spring/summer particularly among females; and extreme weather events (floods, droughts) elevate suicide risk especially in vulnerable groups, older adults, pregnant women, and those with pre-existing mental illness. However, the review exposes major evidence gaps: nearly all studies originate from high-income countries despite LAMICs facing more severe climate impacts, and only 2 of 43 studies examined suicidal ideation while 41 focused exclusively on fatal outcomes, missing the critical early intervention window.

Minor revisions:

(1) Address that all ROBINS-E-assessed studies showed “some concerns” for confounding bias, a limitation affecting causal inference that warrants more explicit discussion.

(2) Briefly acknowledge how excluded qualitative studies could inform intervention design despite valid methodological rationale for exclusion. The policy recommendations (heat-mental health warning systems, cooling centers, eMHPSS in disaster preparedness) are actionable but need more detail on LAMIC implementation given resource constraints.

---

## [Editor Report]

Dear Prof Ahmed,

Thank you for your resubmission. Please kindly address the comments put forth by our reviewer prior to resubmission.

Thank you and all the best,

Dr. Sandersan Onie

---

## [Editor Report]

Dear Prof Ahmed,

Thank you for making the revisions requested. I am now happy to recommend this manuscript for publication.

Congratulations on this important contribution to the literature.

All the best,

Dr. Sandersan Onie